# Next-Generation Pedal: Integration of Sensors in a Braking Pedal for a Full Brake-by-Wire System

**DOI:** 10.3390/s23146345

**Published:** 2023-07-12

**Authors:** Jose Ángel Gumiel, Jon Mabe, Fernando Burguera, Jaime Jiménez, Jon Barruetabeña

**Affiliations:** 1Electronic Technology Department, University of the Basque Country (UPV/EHU), 48013 Bilbao, Spain; jaime.jimenez@ehu.es; 2Fundación Tekniker, 20600 Eibar, Spain; jmabe@tekniker.es; 3BATZ S. Coop., 48140 Igorre, Spain; fburguera@batz.es (F.B.); jbarruetabena@batz.es (J.B.)

**Keywords:** brake-by-wire, pedal brake, functional safety, automotive sensors, reliable systems

## Abstract

This article presents a novel approach to designing and validating a fully electronic braking pedal, addressing the growing integration of electronics in vehicles. With the imminent rise of brake-by-wire (BBW) technology, the brake pedal requires electronification to keep pace with industry advancements. This research explores technologies and features for the next-generation pedal, including low-power consumption electronics, cost-effective sensors, active adjustable pedals, and a retractable pedal for autonomous vehicles. Furthermore, this research brings the benefits of the water injection technique (WIT) as the base for manufacturing plastic pedal brakes towards reducing cost and weight while enhancing torsional stiffness. Communication with original equipment manufacturers (OEMs) has provided valuable insights and feedback, facilitating a productive exchange of ideas. The findings include two sensor prototypes utilizing inductive technology and printed-ink gauges. Significantly, reduced power consumption was achieved in a Hall-effect sensor already in production. Additionally, a functional BBW prototype was developed and validated. This research presents an innovative approach to pedal design that aligns with current electrification trends and autonomous vehicles. It positions the braking pedal as an advanced component that has the potential to redefine industry standards. In summary, this research significantly contributes to the electronic braking pedal technology presenting the critical industry needs that have driven technical studies and progress in the field of sensors, electronics, and materials, highlighting the challenges that component manufacturers will inevitably face in the forthcoming years.

## 1. Introduction

Drive-by-wire is an automotive technology that uses electrical or electro-mechanical systems to perform vehicle functions, avoiding mechanical linkages. New trends in the automotive industry are accelerating the adoption of X-by-wire technology, such as electrification and the use of autonomous vehicles [1,2]. This approach offers several advantages, including minimizing the number of mechanical parts and connections, reducing the overall weight, and providing a faster response time than traditional systems [3].

The present article outlines a concept for the next generation of braking pedals. An element that moves away from its commoditization and is renewed with the introduction of novel features. The main attribute of this pedal is its brake-by-wire (BBW) architecture. However, other innovative elements are also proposed, with electronics being the main actor in this transformation. A pure BBW system would eliminate the need for hydraulics by using motors to actuate the brake calipers, unlike current technology [4] where the system is designed to provide braking effort by building up hydraulic pressure in the brake lines.

The research aims to examine what the pedal in cars in coming years might look like and which additional features it might introduce. The central hypothesis posits that BBW technology will become more prevalent in the automotive industry [5], as explained in Section 2, due to the increasing use of electronic systems. Section 3 depicts how several OEMs were consulted to obtain research feedback, highlighting the attempt to accomplish certain points of the ISO 26262 regulation. An important aspect of this research is to address the challenge of developing high-quality components at low cost, which is a pressure exerted by original equipment manufacturers (OEMs) in the automotive sector. Considering that electronics dominate the price of the whole pedal, it is essential to find safe, reliable, accurate, and low-cost sensors. Section 4 describes the concept of the next-generation braking pedal. This study also considers current trends focusing on lightweight components, power efficiency, and differentiation concerning brake pedal design. In addition, the research explores new functionalities, such as retractable pedals, which could enhance the capabilities of future brake pedals, particularly considering the trend toward autonomous vehicles. Finally, the conclusions of this work are addressed in Section 5.

This research was conducted at the Advanced Engineering Area of BATZ, an established automotive supplier headquartered in Northern Spain. It resulted in the development of a fully functional BBW pedal with unique features. This project was a result of speculation by BATZ on the future of pedal boxes, specifically brake pedals, when the market was not yet clear on the path to be taken. However, there was a clear tendency to integrate more technology in the form of sensors and actuators [6]. During this project, BATZ gained knowledge of the standards and technologies to be included in future brake pedals. At the same time, open discussions with various OEMs allowed BATZ to become an effective partner when these OEMs were internally looking for a new paradigm of braking in new generations of vehicles. Aspects such as pedal feel and travel must be studied from an ergonomic point of view rather than from a mechanical performance perspective when the pedal box is not mechanically coupled to the vehicle’s braking system. This proactive stance prepares the company for the industry’s upcoming challenges, such as electronic integration, economic competitiveness, and product differentiation. This results in innovative, competitive, safe, reliable, and high-value-added components.

This investigation is important because it opens new paths toward BBW and the electronic brake pedal (eBrake). One of the problems with the proposed technologies is that they are very innovative, and there was, and still is, little information available about them. Also, the problem that there is currently no eBrake system in production makes it difficult to see the needs of the different OEMs when even they are currently investigating the possible requirements to be addressed. This is because conventional brake pedals are limited in feel and performance by the vacuum booster to which they are attached. But designing a new paradigm of eBrake, without any mechanical linkage to the rest of the braking system, frees the pedal from this dependency [7]. Therefore, certain OEMs have tried to replicate the feel of the traditional pedal boxes, while others are looking for a new model with reduced travel and unique ergonomics. A clear standard solution is yet to emerge.

This work has culminated in the construction of various BBW pedal systems with innovative features, such as a variable feel simulator and a retractable mechanism. Furthermore, two novel sensors have been developed, and the energy consumption of an existing sensor in production has been improved. Lastly, we have successfully manufactured the first affordable plastic brake pedal using WIT.

## 2. State of the Art

Integrating electronic components into the vehicle is one of the major trends in the automotive industry [8]. Electronic systems are becoming increasingly common, and many components are being transformed so that electronics provide new features and functionality [6]. This affects everything from rear-view mirrors [9] and sun visors [10] to the new advanced driving assistance systems (ADASs). Every component has room for improvement, no matter how simple or what the safety implications.

Factors such as the enhancement of vehicle control systems, the advent of electric mobility, or the connected car receiving over-the-air (OTA) updates have accelerated the drive-by-wire roll-out. This technology replaces traditional mechanical components with electro-mechanical systems by integrating sensors and actuators [3]. In addition, it offers benefits such as weight and size reduction and improved efficiency in dynamic vehicle control [11].

Drive-by-wire is a technology that comes from the aeronautical industry. The first by-wire system to be integrated into a commercial car was the throttle-by-wire or electronic throttle control (ETC), which appeared in the 1988 BMW 7 Series [12]. Since then, the use of the electronic throttle has become widespread, and the traditional throttle, operated by a Bowden cable, is now practically considered obsolete [1]. This system uses sensors to measure pedal travel; the electronic control unit (ECU) receives the data, which manage the required fuel and optimal air intake, sending commands to actuators. ETC eliminates the mechanical linkages simplifying the system, allowing an optimal fuel-air mixture for each situation, allowing data exchange with other systems (such as cruise control or traction control), and reducing pollutant emissions [13]. The proliferation of electronic control systems paves the way for other elements that have acquired drive-by-wire capabilities, such as steer-by-wire (SBW), used for the first time in the 2014 Infiniti Q50 [14], or brake-by-wire (BBW). This system will be discussed in detail in this article, although these are uncommon in road vehicles at the moment.

BBW represents a system without any linkage between the brake pedal and the wheel brakes. One of the pioneers to use this term was Keith Holding, the Chief Development Engineer of the Advanced Engineering Department at Lucas Automotive. In 1990, he highlighted the potential benefits of BBW [15]. Keith hypothesized that the overlooked data at that time could enhance vehicle deceleration, with a computer determining the optimal braking distribution based on the road conditions and the driver’s braking force input [16]. In 1996, Bosch was also working on an electro-hydraulic braking system with similar objectives, aiming for a more efficient distribution of braking force [17]. However, it took a decade from the inception of the BBW concept to its initial integration into a road vehicle.

The earliest appearance of this system was in 2001 in the Toyota Previa/Estima [18]. At that time, the Japanese company had just started manufacturing hybrid vehicles, and they quickly understood that BBW technology could combine efficient braking with superior handling and safety. This technology was called the electronically controlled brake (ECB) system [19]. They combined various data from different parts, such as the pedal stroke sensor, speed sensor, master cylinder and wheel cylinders sensors, or yaw sensor, among others. This information was used to calculate the optimum hydraulic pressure [20]. Since this vehicle also had a regenerative brake system (RBS), there was communication between the brake control and the hybrid control. This allowed for the maximization of kinetic energy collection from braking, which could then be converted into electrical power. It could even recover energy from each wheel [21]. Three years later, this system was integrated by Lexus, Toyota’s luxury vehicle division, into the RX400h model [22].

Another vehicle using this technology was the Ford Escape Hybrid [23], which went into production in 2004. Although Ford claimed to have independently developed its hybrid powertrain technologies without external assistance, their similarities to Toyota’s patented technologies required Ford to obtain licenses to use them [24].

Although Toyota and Ford introduced this technology without issues, the same cannot be said for Daimler-Chrysler. At the turn of the millennium, Mercedes-Benz relied on the sensotronic brake control (SBC) system [19], which was developed in cooperation with Bosch [25,26]. It looked promising and was introduced into Daimler’s top vehicles. With fine-grained control of pressure at each wheel, SBC offered a unique platform to implement skid protection and traction control. However, it also included other features, such as the distribution of braking force in each wheel during cornering, maximum stopping power, and drying of the brake discs if a film of water formed on them. It even managed to eliminate the vibration associated with the conventional anti-lock braking system (ABS) during hard braking [25,27]. The technology for the time when the system was born seemed like a great milestone in braking technology; however, the execution was imperfect [28], and they discontinued this system [29].

There were certain problems with the SBC braking system; it was not robust enough, resulting in premature deterioration, causing the system to fail and stop working [28]. Nevertheless, this was not a hazard to the driver, as the vehicles had a backup hydraulic-only braking system. In the event of electronic failure, SBC reverts to a hydraulic master cylinder. The downside is that the pedal feel and the sudden loss of stopping power can disconcert the driver and increase the stopping distance. Safety is also applied in cases of non-electronic failure. If there is a pump failure, a high-pressure reservoir exists capable of stopping the vehicle electronically.

Failures of the SBC systems led to the recall of over two million Mercedes models, primarily the E-Class. Despite this, the brand issued a 25-year warranty from the date of purchase for those cars that integrated the system [30].

On the other hand, Formula One, the highest class of international open-wheel single-seater formula racing, has influenced the development and use of BBW systems. In 2014, the ERS-K system, which is responsible for recovering energy from braking and storing it in an electric battery, became mandatory [19]. The aim was to increase the vehicle’s energy efficiency and use energy that would otherwise be wasted or dissipated as heat. When the driver reaches a corner and brakes, three variables come into play: the braking system itself, the engine brake, and the ERS-K system, which does not always behave in the same way. This requires much greater variations in rear-wheel braking torque than before and can make it difficult for the rider to control the vehicle [19].

The ERS-K system made “engine braking” much more effective but added uneven energy harvesting, which affected the vehicle’s controllability [19]. This led to the incorporation of BBW systems, which detect the desired braking power input from the driver and modulate the force applied to the brake discs based on the available assistance from the energy recovery systems.

The FIA have allowed the use of electronically controlled brake balances [31]. These systems regulate the force applied to the rear brakes, ensuring harmonious operation with the energy recovery system during deceleration and harvesting. This compensation is necessary to counteract the considerable influence of ERS-K on brake balance and braking stability. BBW also require a feel simulator to give the driver feedback on the vehicle. As a backup, the driver’s pedal is still hydraulically connected to the rear calipers, and braking must be possible even if the electronic brake-by-wire system fails [19].

Fast forward to the present; in 2016, Continental announced the MK C1 [32], a BBW technology designed for road vehicles. It consists of an electro-hydraulic system capable of generating a high braking pressure at high speed while maintaining control of the car, thus preventing accidents and protecting pedestrians. The supplier announced it as an enabler for highly automated driving, but it also has other benefits in ordinary driving. It enables energy harvesting from the braking force, reduces 30% of the weight of a traditional braking system, and provides efficient braking dynamics. Furthermore, it contributes to reducing CO_2_ emissions [33]. This system integrates control systems, such as ABS and electronic stability control (ESC), in the same compact module.

Thus far, it is known that two OEMs have introduced this system in their vehicles. It is present on the Alfa Romeo Giulia and Stelvio [34,35] models and in the Audi e-tron [36].

Alfa Romeo have taken additional advantage of this system apart from weight reduction and safety features. As a sporty brand, BBW technology allows them to fine-tune the pedal feel simulator to suit various driving scenarios. It enables the setting of a less aggressive pedal feel for relaxed driving in traffic or a more aggressive response for track conditions. This enhanced adaptability of the system expands its capabilities and usability.

The Audi e-tron quattro is equipped with the MK C1 system, integrated with a feel simulator and a displacement sensor. The sensor detects the level of deceleration desired by the driver. This information is then transmitted to an ECU, determining the best braking mode: regenerative, hydraulic, or a combination of both. Audi claims that regenerative braking is sufficient for all braking maneuvers with a deceleration of less than 0.3 g [36].

This system normally has open hydraulic valves between the master cylinder and the hydraulic circuits. Once the system is powered up and has passed its internal diagnostic tests, the brakes are switched to “by-wire mode,” where the brake pedal is decoupled from the hydraulic circuits by closing the valves: oil displaced from the master cylinder is diverted to the pedal feel simulator. Pedal travel and pressure are measured, and the actual brake pressure is generated by an electric motor inside the brake controller. Those valves will stay open if this controller is unpowered or has a fault condition. In this case, the brake pedal will directly actuate the hydraulic brake circuits. The pedal will have an abnormally long travel and feel very spongy, but the vehicle will be able to brake.

The evolution of Continental’s MK C1 is the MK C2 [37], which can be attached to either a conventional or an electronic pedal. This opens a window for electronic pedals (ePedals) like the one proposed in this article and leads to the assumption that Continental will be one of many suppliers working on such systems. Traditional pedals will have to evolve.

Today, the use of BBW systems in road vehicles is not yet mainstream. Only a few models, as shown in Table 1, are equipped with this technology, and these tend to be high-end cars. With the advent of hybrid vehicles and electrification, there is a need for systems that optimize energy efficiency; thus, the demand for BBW systems is expected to increase. BATZ have already designed and produced pedals that can be attached to boosters like those mentioned above; the company is now receiving requests to develop ePedals. The research and development work presented below has lasted two years and has served to understand the needs of OEMs beforehand, analyze state-of-the-art technologies, and develop functional prototypes while investigating novel technologies. This learning experience puts the company in a favorable scenario to respond to these requests. The eBrake is an emerging reality that will improve vehicle performance, reduce reaction and braking times, and increase safety for the driver and other road users while simplifying communication between the vehicle’s multiple dynamic and motion control ECUs.

## 3. Methodology

In a previous study, a comprehensive analysis was conducted on the evolution of various automotive components and their increasing integration of electronic technologies. This analysis concluded that electronic innovation will permeate every vehicle component [6], including brake pedals [38]. Consequently, a research and innovation (R&D) strategy has been pursued to remain state-of-the-art in BBW technology, ensuring alignment with the latest advancements in the field.

The development of this project has been led by the BATZ Group, a corporation with a strong mechanical engineering foundation currently transitioning toward mechatronics. Based on an extensive study conducted on effective strategies to incorporate electronic knowledge into traditional automotive Tier 1 companies [39], it was determined that organic growth was the most appropriate approach for BATZ. Aware of the latest market trends, the company initiated this project within the Advanced Engineering Department, a specialized team dedicated to research and development. This team drives organic growth by exploring new technologies and products within the company and fosters employee engagement and motivation, encouraging the acquisition of new skills and knowledge.

Research allows the generation of knowledge within the company, which leads to acquiring experience, increasing competitiveness, and facilitating the approach to OEMs through innovative solutions.

The introduction of electronics in a safety-critical element, such as the brake pedal, must comply with the ISO 26262 regulation. Although this is not the development of a commercial product, this international standard will be partially followed to become familiar with its application.

The project’s first phase was based on ISO 26262-3(5), corresponding to Item Definition. To define the brake pedal, the engineering team met and began brainstorming ideas, exploring innovative approaches to obtain representative data to characterize the operation of a conventional pedal electronically. Due to the novelty of the project and the lack of previous experience, the initial study was quite limited; therefore, several OEMs were contacted to obtain a more accurate vision and understanding of their needs and reality.

The main intention was to start working with by-wire technology so that learning would allow the company to offer customized electronic pedals to OEMs. This required a state-of-the-art study and an understanding of customer needs and requirements. At this stage, discussions with the vehicle manufacturers were enriching, as they provided information that the engineering team was unaware of or had overlooked. In addition, they shared their interests in some of the features they were looking for in such a system and those that had already been considered in the design and that they saw as potentially interesting.

The feedback from the OEMs was constructive and allowed the engineers to focus on the main objectives of the project, as well as to direct their research to define the implementation of the system. In this process, an extensive analysis of scientific articles, patents, and doctoral theses was undertaken to gather valuable insights.

Once the item functionalities are clear and well-specified, the next step, according to ISO 26262 [40], is hazard analysis and risk assessment (HARA). In this section, the potential hazards of each item’s function are analyzed, evaluating the hazardous event, under which conditions it can occur, and its consequences. Each hazard is classified according to three criteria: severity, exposure, and controllability. The combination of these three determines the ASIL level, and one or more safety goals are identified to prevent the hazard from occurring.

Based on this information, the next phase is related to the project’s development according to the item functions and the safety concept. ISO 26262 uses a V-model development process, demonstrating the relationships between each phase of the development life cycle and its associated testing phase. The engineering team was aware of this model. Regardless, this standard was only partially applied in this research project, emphasizing areas where knowledge needs to be acquired and which are most significant to rapid business development. Although the norm is for electronic development only, mechanical design and pedal construction will also occur at this stage.

The electronic components are carefully chosen and designed in-house according to the ASIL standards. The optimization of power consumption has been achieved for an existing sensor, which has been benchmarked. Additionally, two innovative prototypes have been manufactured, including a novel printed-ink technology force sensor. Its repeatability and linearity have been analyzed. Nevertheless, it requires further testing for its feasibility in automotive applications. The next step is to subject the sensors to rigorous climatic and durability tests to evaluate their performance and reliability.

All this work has resulted in various prototypes of the BBW pedal. The difference between the prototypes lies in the features to be tested, such as the passive and active feel simulators or the retractable mechanism. These prototypes all integrate sensors to measure exerted force and angular displacement since they inherit the traditional shape of the pedal. They are functional and have been validated through repetitive cycle testing. In addition, they offer the potential for further improvement by reducing the number of mechanical parts and simplifying the design. In parallel, during this research project, a team specialized in plastics have developed the first affordable plastic brake pedal, which successfully performed the collapse test.

## 4. BATZ’s Next-Generation Pedal

BATZ is a global automotive supplier based in Igorre, Northern Spain, part of the Mondragon Corporation. The company specializes in metal and plastic stamping and is expanding into electronic components, integrating them into its product portfolio. BATZ is involved in two activities, tooling and systems, with systems consisting of mechatronics on the one hand and lightweighting and active aerodynamics divisions on the other.

The research project presented in this article is an electronic brake pedal that enables the integration of a full BBW system in a vehicle. The name for this project is NG-PED (an acronym for next-generation pedal). It consists of a brake pedal with the necessary sensor technology to send its position to the ECU so the vehicle can perform accordingly. In addition, other electronic features for the pedal of the future have been studied and tested, such as an active feel simulator using shape-memory alloy (SMA) materials, as well as non-electronic features, such as a passive feel simulator and the manufacture of an all-plastic pedal molded using water-injection technology (WIT). Introducing such pedals could lead to weight savings or a reduction in braking time, as well as support for other systems, such as energy harvesting, or the optimization of different elements, such as ABS or ESC [41].

In the current era of highly connected vehicles, integrating advanced sensor technology plays a pivotal role. Vehicular sensors collect valuable data, enabling the optimization of the vehicle’s control systems and enhancing the reliability, robustness, and accuracy of its dynamic behavior. By-wire systems, such as the one proposed, are a prominent feature of modern automotive technology that facilitates real-time adjustments during driving, eliminating the need for driver intervention and seamlessly enhancing the overall driving experience [42].

### 4.1. System Definition

As explained in the methodology section, the first step is to define the overall system and its functions. Subsequently, this section will outline the proposed features to be introduced in the pedal and the requirements for achieving a robust, safe, and lightweight system. These requirements are derived from the hazard analysis; only the most relevant ones will be mentioned here.

The primary goal of this project is to develop a BBW pedal system. It will be mandatory to include electronic sensors capable of measuring physical magnitudes associated with an action on the pedal. Based on the hazard analysis, the team has determined that at least two sensors of different natures must be incorporated. Different sources should power these sensors. The braking system is critical and any failure could pose a life-threatening risk; it was therefore determined that the electronic system must meet the highest level of safety, ASIL D. Table 2 presents a partial hazard analysis that led to the identification of these requirements based on the ASIL classification.

This table provides an illustrative example of hazards that can affect the pedal system. It should be noted that the list of hazards is more extensive and comprehensive. The next step is determining the ASIL classification for these hazards and establishing the corresponding safety goals. The examples presented in Table 3 serve to demonstrate this process.

Based on this information, the next step is to define the requirements to accomplish the safety goals. It is important to note that the level of completeness and comprehensiveness of the HARA directly impacts the system’s overall safety. The higher the level of completeness and comprehensiveness of the HARA is, the safer the system will be.

From the previous tables, individual requirements are derived for each function, aligning them with the specified safety goals. To illustrate this with examples, safety goals derived from H.1 and H.2 can be examined. Incorporating two distinct sensors from different manufacturers is necessary to meet these requirements. These sensors should measure different physical magnitudes and be positioned on separate printed circuit boards (PCBs) in different pedal areas. Another essential requirement, derived from H.3, entails the inclusion of two independent connectors within the pedal system. Additionally, the electronics should be designed to allow the OEM to power them using different power sources, ensuring fault tolerance [43].

In addition to the functional safety requirements, other system requirements may be demanded by drivers or OEMs or imposed by Tier 1 itself to be more competitive. For example, in the case of a BBW, the drivers prefer to ignore that there is no physical connection between the pedal and the brakes, so the pedal feel should be as close as possible to that of a conventional pedal [44]. In this case, it is necessary to develop a feel simulator to characterize the pedal displacement and the exerted force. On the other hand, OEMs pay special attention to the component’s price. The automotive industry is an extremely competitive industry, and a small cost reduction on a component can mean a large profit on large volumes. One requirement could be to use manufacturing techniques that reduce the purchase price. To be competitive today, developing products with high-added value is necessary. Offering lightweight solutions or energy-efficient designs can put the supplier in an advantageous position, whether through innovative manufacturing techniques, such as WIT, or by developing electronics with energy-efficient modes.

### 4.2. Brake-by-Wire: The Mechatronic Pedal

The subsequent phase involves developing the system design after establishing the pedal system requirements and outlining the functional safety requirements.

Based on the previous analysis, the mechatronic pedal must have sensors that measure the physical magnitude associated with the pedal position. There are multiple ways to detect force exerted by the driver on the pedal. Sensors commonly used in the EMB actuator include the following types: pressure, torque, angular displacement, and axial displacement sensors [45,46]. In automotive applications, choosing the right sensor involves considering factors such as resistance to moisture, dust, oil, and thermal drift. Desired features include excellent linearity, wide operating range, cost-effectiveness, high resolution, contactless rotation, low power consumption, ease of manufacturing, and simple signal conditioning. Capacitive rotary and optical sensors, although highly sensitive, have been excluded due to their dependency on dry and clean environmental conditions [47]. The proposal for this project is to use a combination of two different sensors: an angle sensor and a force sensor. The figure below is an illustration of how this system would work.

On the left side of Figure 1 is the pedal system. It consists of two sensors that measure the force applied by the driver and the pedal position based on the rotation angle. The force sensor has a single output, while the angle sensor generates two independent signals. In the middle is one of the several electronic control units (ECUs) within the vehicle [48]. This system receives signals from the sensors. It operates on the braking system, either by generating hydraulic pressure or controlling a set of actuators that act on the brake calipers. It is important to note that in the current state of BBW technology, production is limited to electrohydraulic braking systems, commonly referred to as “wet” systems. Conversely, actuator-based technology, known as “dry” systems, poses certain challenges and remains an ongoing research subject, with only prototype implementations currently available. The ECU may also use the pedal input data and communicate with other braking-related subsystems to make the vehicle more controllable and maneuverable, improving kinematics and dynamics. Introducing this type of sensing technology can be an enabler for autonomous driving since there is no need for human intervention to actuate over a vehicular system explicitly; the ECU has this capability based on the information received from the multiple sensors [49]. 

This project implements only the pedal part, which senses the driver’s activity and transmits it to the vehicle through electrical signals.

Considering the functional safety requirements for achieving an ASIL D classification, the pedal system has been equipped with a combination of two ASIL B sensors. There are multiple approaches to meet the ASIL D requirements, as shown in Figure 2, and this combination is one of them.

Figure 2 shows the ASIL decomposition according to ISO 26262. In this case, and based on the figure, it is possible to achieve the required redundancy for an ASIL D system by using two ASIL B qualified sensors. The following subsections will provide further details on the sensor selection process.

#### 4.2.1. Angle/Travelling Sensor

The conventional pedal mechanism facilitates the conversion of the force exerted by the driver into a rotational motion. Integrating electronic components into the pedal design necessitates an efficient method to detect and measure the driver’s braking inputs. BATZ have demonstrated their expertise in this domain by successfully implementing a secondary brake sensor with Hall-effect technology currently in production. However, this research investigates the feasibility of utilizing inductive technology, considering its accuracy, noise immunity, and cost-effectiveness advantages. Furthermore, energy-efficient strategies are explored to optimize the power consumption associated with the Hall-effect sensor.

Inductive sensors, which operate without physical contact, measure the perturbation of a magnetic field caused by a conductive target. In this type of sensor, a coil generates a high-frequency magnetic field. A current is induced within the object as a metal object approaches the changing magnetic field. The inductive sensor utilizes an oscillator to drive the coil, and when a metal object nears the coil, it alters the inductance, resulting in a frequency or current change. These variations are then detected by the trigger circuit, enabling the determination of the target’s position. Notably, the system requires the presence of two coils positioned at different physical locations to establish a reference for position detection and enable accurate measurements [50].

One advantage of inductive sensors is that any metallic target will suffice. On the other hand, Hall-effect sensors and magnetic sensors require a permanent magnet. A metallic target is a more cost-effective solution, and in addition, it is also more robust. Magnetic strength can be affected by environmental changes, such as temperature, depending on the composition of the magnet. Meanwhile, inductive sensors excel at elevated temperatures, whereas other systems struggle. They are also accurate, with errors below ±0.1% over the full measurement range at room temperature and below ±0.3% when there is higher temperature or air-gap variations between the target and the sensor.

In the present scenario, where electrification is becoming more and more widespread, inductive sensors have an advantage over magnetic sensors. This technology is immune to magnetic fields and systems such as electric steering or electric motors, which generate stray magnetic fields.

Another reason for choosing an inductive sensor is its freedom in determining the type of motion to be measured and the design of the PCB form factor. In this case, an arc-shaped displacement was expected to measure an angular position.

Price was not a determining factor; however, it is common for inductive sensors to be more expensive than magnetic or Hall-effect sensors [51]. Nonetheless, the cost difference is counterbalanced by using a metal target instead of a magnet.

Considering the abovementioned factors, an inductive sensor was specifically designed and manufactured for this project. All the electronic components, including passive semiconductors, were qualified for automotive applications. The sensor was rated with an ASIL B classification from the semiconductor manufacturer, meeting the required safety standards.

Specific communication protocols are employed in the automotive sector based on data rate, fault tolerance, and cost. The protocol for this application is the SAE J2716 standard, commonly known as single edge nibble transmission (SENT). This digital signal transmission protocol utilizes a single wire for data transmission. It operates one-way, continuously transmitting data from the sensor to the ECU without requiring request commands [52].

The adoption of the SENT protocol offers several advantages. Firstly, it reduces costs as it enables point-to-point communication. Additionally, it maintains a high transmission accuracy and speed, supporting rates of up to 30 kbps [53]. The protocol also incorporates fault tolerance by including a cyclic redundancy check (CRC) at the end of each frame, allowing the ECU to verify the integrity of the received signal.

In this project, the energy efficiency of the sensors was also considered. With electrification, OEMs pay more attention to the vehicle’s power consumption. Although a single sensor may not have excessive power consumption, this can change drastically when there are around a hundred sensors in a modern car [54]. If electronic systems are not designed with efficiency in mind, when they are in the vehicle, there will be a significant waste of energy [55,56]. For this reason, certain sensors are equipped with a sleep-mode function, such as the Hall-effect sensor we integrate.

Sleep mode is a feature that allows the sensor to enter a low-power state [55]. However, the implementation of this power policy can vary depending on the specific architecture. There are different approaches taken by OEMs in handling sleep mode for the sensors. Certain OEMs only set the redundant sensors to sleep mode while keeping the primary sensors active. Others may disable the pedal system’s redundant sensors and set the primary sensors to sleep mode. Alternatively, certain OEMs may require a specific signal to be received, triggering power-saving measures for the ECUs. In certain cases, OEMs may even mandate bi-directional communication between the pedal system and the ECUs, allowing both to enter sleep mode and conserve power during periods of inactivity.

The proposal presented in this article is an energy-efficient Hall-effect sensor. It has a switch-type signal that informs the ECU of its state (pressed or released). When idle, the ECU can send a signal to set the sensor into sleep mode, resulting in very low power consumption. The sensor continuously checks whether the driver has pressed the pedal or not. The moment the pedal is pushed, the switch signal changes, and the ECU must detect this change and wake up the sensor. When the sensor wakes up, it starts sending SENT frames indicating the current position of the pedal.

The following graph, Figure 3, shows a comparison of the power consumption of the sensors. On one side is the Hall-effect sensor with sleep function enabled and disabled, including a magnetic sensor from another competitor. On the other side is the inductive angle sensor. Finally, the power consumption of the printed ink-based sensor, a prototype developed in this project (which will be addressed later), is also shown in the graph. All the tests were performed to measure the power consumption in both scenarios: when the pedal is idle and when it is active.

#### 4.2.2. Force Sensor

A force sensor changes its properties when a force, pressure, or tension is applied. This is a desirable choice for an electronic brake pedal because the driver displaces the pedal by exerting a force on it. While a traditional pedal design relies on a spring and hydraulic pressure for providing feedback on braking force and vehicle response, an electronic brake pedal equipped with a force sensor offers enhanced capabilities for measuring and transmitting precise force information.

It should be considered that the traditional brake pedal exhibits movement mainly because these systems function as a hydraulic pump, where the movement of the pedal itself is necessary to create a flow in the hydraulic fluid to stop the vehicle. But this paradigm is changing radically with the introduction of fully electronic BBW systems, where the driver pushes no fluid. Thus, if we forget the inheritance of traditional braking systems, using force sensors, even with the possibility of zero displacement, should be the main factor.

A commercial force sensor was chosen for this prototype, although research was also conducted on strain gauges, which will be discussed subsequently. Opting for the force sensor provided multiple advantages, including a reduction in development time. Firstly, the sensor already integrated a Wheatstone bridge [57,58], making it a compact solution. Additionally, the sensor offered a comprehensible analog output, which is advantageous as it aligns with the widespread practice of integrating analog-to-digital converters (ADCs) into the vehicle’s ECUs. Furthermore, having two different signals in the pedal system is beneficial for functional safety, providing redundancy through different protocols [59]. Lastly, the force sensor exhibited a faster response time than the existing position sensor. However, the main drawback of force sensors is their higher price, which is a handicap and poses a challenge in cost-sensitive automotive applications.

There are diverse technologies in force sensing, but a load cell was chosen for this approach. This type of component is essentially a transducer that measures deformations produced by force or weight and converts them into an electrical signal that can be easily measured and analyzed [58]. In the NG-PED design, the force is exerted on the sensor in a downward direction, so the sensor element is more specifically a compression load cell. They are usually made of metal and are designed to be durable and reliable, which is an advantage due to automotive constraints. On the other hand, they can also be expensive compared to other types of force sensors due to the materials and manufacturing processes used in their construction. This type of sensor requires signal conditioning, in which case the electronic circuitry is integrated into the load cell.

Compression load cells can be based on strain gauges, a technology addressed in the next subsection. Regardless, they can also be based on other principles, such as hydraulic or piezoelectric elements. While strain gauge-based load cells are commonly used due to their ability to convert strain into electrical signals, hydraulic load cells utilize fluid pressure for force measurement. On the other hand, piezoelectric load cells employ the piezoelectric effect to generate electrical charges when subjected to mechanical stress. Each type of load cell offers unique advantages in terms of accuracy, response time, and suitability for different applications.

Force sensors generally have a high price. Multiple factors contribute to this: First, the technology used can be complex and difficult to manufacture. Second, the materials used, such as high-grade steel and ceramics, can be expensive. Finally, the performance required for demanding applications, such as accuracy or resolution, can increase costs.

In this project, the sensor would be subjected to high loads, so the ruggedness to withstand fatigue stresses and the elasticity to avoid deformation were two of the main requirements. In addition, an automotive grade was another requirement.

#### 4.2.3. Strain Gauges

As mentioned earlier, pricing is especially important in the automotive industry. A small saving on a part can have a significant impact when manufactured in large batches. For this reason, an attempt was made during this research to find an affordable alternative to the force sensor [60]. In pursuit of this goal, the research identified strain gauges as the most viable approach. Consequently, proof-of-concept testing was initiated to explore the feasibility of employing this technology.

A strain gauge is a sensor that exhibits changes in its electrical resistance in response to an applied force. It converts force, pressure, tension, or weight into corresponding changes in electrical resistance. These changes are measured using a Wheatstone bridge circuit. When external forces act upon a stationary object, they induce two important effects: stress and strain. Stress refers to the internal resisting forces within the object, while strain denotes the resulting displacement and deformation that occurs [61]. Figure 4 illustrates the physical response of a strain gauge when subjected to an applied force.

On the left is the strain gauge at rest. Notably, the length of the wire is shorter and wider than on the right. When a force is exerted on the strain gauge, it undergoes a stretching effect within its elastic limits, preventing any damage or permanent deformation. As a result, the gauge experiences a change in its dimensions, becoming narrower and longer. This alteration leads to an increase in the electrical resistance across its entire length.

To test the behavior of strain gauges for an automotive application, a PCB was designed with a plate in the center that allows a slight movement when a force is applied to that point. The strain gauges are placed at the beginning of this moving part of the board to measure the force. The movement is short and limited because excessive force could cause the PCB to break, rendering the gauges useless.

In addition to the gauges, the PCB contains the electronics necessary to measure the resistivity of the gauges. It also includes an ADC to obtain the digital homolog value and an integrated circuit to read the data and convert them to SENT frames, with the corresponding filtering of the signal output.

A universal testing machine (UTM) analyzed this prototype’s behavior. This is a machine to test the compressive strength of materials. In this scenario, this tool is used to apply a controlled force and vertical displacement across the PCB while recording the test and the SENT signal from the sensor. The graph in Figure 5 shows both values.

The test involved applying a force ranging from 0 N to 1000 N to the sensor, resulting in a displacement of 0.6 mm on the PCB plate. To counteract this displacement, a coil spring is positioned beneath the PCB. This spring compensates for the sensor movement and contributes to the curved shape observed in the sensor readings, which vary based on the sensor’s displacement. Initially, the slope of the curve is relatively flat, gradually becoming steeper as the displacement increases. The sensor has been configured with an offset of 400 LSB.

Although the first impressions were positive, this approach had an inherent problem: the manufacturing processes needed to be more suitable for a mass-produced automotive product. The reason was that the gauges were attached to the PCB with an adhesive and soldered to the pads with tin using a copper wire that protruded from the component. While this assembly method sufficed for a prototype and proof of concept, it can only be brought to the market with a remarkably high economic impact due to the manufacturing processes.

The difference in elastic moduli between the adhesive and the strain gauge substrate impacts the maximum achievable gauge factor of the final sensor stack, and the process itself can introduce complications, as mentioned above [62]. However, this study proposes a method that addresses these challenges by directly printing the strain gauges onto the PCB. Inkjet-printed flexible strain gauges typically use composite inks consisting of conductive particles embedded in either a less conductive or insulating polymer matrix. Since the conductive particles are much more conductive than the polymer matrix, the conductivity of the printed material is strongly dependent on the particle loading, particle size, and distribution, which is affected by strain [60]. Traditional strain gauges involve relatively expensive fabrication processes, unlike additive print-manufactured devices [63]. This provides an ideal opportunity to evaluate the viability of printed-ink strain gauges in the automotive sector, where pricing plays a crucial role. To date, a dozen prototypes have been fabricated, as shown in Figure 6, and certain data have been collected. The collected results are presented below in Table 4, although further research is needed to analyze whether the proposed solution is applicable in the automotive industry.

The behavior of the prototypes was analyzed as shown in Figure 7 yielding the following observations: Firstly, high resistance values (kΩ) were discovered, exhibiting significant variation among them. This caused imbalances between the arms of the bridge, resulting in a considerable disparity in offset levels across the manufactured prototypes. Secondly, the excellent linearity exhibited in response to the applied force was a notable characteristic. It is worth noting that the output range surpassed that achieved with the previous design.

Thus far, more research is needed to ascertain whether this solution is valid for automotive applications. The investigation will be followed by integrating these bridges with a conditioner and other electronics. If the preliminary results are satisfactory, endurance and environmental tests will be performed.

#### 4.2.4. Other Technologies

The pedal of the future can be an element with high added value. In this project, research has been carried out to develop a customizable pedal hardness mechanism that caters to individual driver preferences. A mechanism with three positions has been designed to provide a neutral feel. Regardless, it can be configured to have a softer feel for urban driving or a stiffer setting to enhance the sporty driving experience.

This is achieved using tensors from shape-memory alloy (SMA) materials [64]. These filaments are heated by an electronic controller applying an electrical current, making them contract and move a mechanism that locks the pedal in the selected positions. SMA filaments have been successfully tested for the active feel simulator, but there are concerns about the suitability of SMA for the automotive industry. These filaments are extremely sensitive to temperature and friction, which adds complexity to the controller [65]. However, this feature is a bit premature, as there is currently no production of an adjustable passive feel simulator. Thus, the need for an active system is not yet on the horizon.

Another trend in the automotive industry is the use of lightweight components. OEMs want to reduce the weight; the main reason is to increase the efficiency of the vehicle, whether it is an internal combustion engine or an electric motor vehicle. Brake pedals are critical car parts that require high fatigue and impact resistance. Traditionally, these pedals were crafted from metal; however, advancements in manufacturing processes and materials have made it possible to achieve the desired robust performance using plastics. The benefits are a lighter weight, more flexible design, component consolidation, and cost reduction.

In the research, BATZ has developed a full plastic pedal through a water injection technique (WIT) process. BATZ was one of the very first companies to introduce this technology on clutches, and it is present in vehicles such as the Mini (F54, F55, F56, F57, and F60), the BMW 1 Series (F52 and F40), and the BMW 2 Series (F44, F45, F46, and U06), among others. WIT is still a minor technology and is only in production in clutch pedals. Using it in brake pedals requires different materials and is a step forward from the previous state of the art. The obtained result is a closed-section pedal that resists torsional stress exceptionally well. The pedal’s inside is hollow, as shown in Figure 8, which makes it much stiffer and harder in extreme conditions such as panic braking, saving the required amount of plastic and reducing the weight.

This pedal is based on a real production unit (Volkswagen Golf, Left Hand Drive (LHD), Automatic), and an effective weight reduction of 45% compared to the original was achieved. The reason for manufacturing this pedal with WIT technology is that, due to its geometry, it must withstand significant torsional stress. This is because of the high geometric lateral displacement of the plate with respect to the axle.

The prototype has been manufactured using an unusual sustainable material: polyethylene terephthalate (PET) composite material reinforced with recycled carbon fiber (rCF), with a fiber volume fraction of 30%. To test the resistance of this pedal, several prototypes were created and subjected to collapse in three different scenarios. The collapse load has been applied on both edges of the pedal plate (left/right) at room temperature (RT) and, in the most extreme case (left), at 80 °C. The results showed a consistent and impressive performance in terms of stiffness and strength. Figure 9 depicts the collected data from the tests, accompanied by a sketch illustrating the shape of the pedal and the points where the load was applied.

When testing pedals, the most favorable scenario occurs when the force application point is aligned with the pedal axis. This prototype has an offset between the pedal plate (and the force application points) and the axis, with a more pronounced offset on the left side. The collapse test performed at room temperature with force applied on the right side showed a resistance of just over 3000 N. In contrast, when the force was replicated on the left side, a slightly higher deformation was observed, leading to failure at 2800 N. Notably, at elevated temperatures, the deformation of the pedal increased, causing failure at a higher force threshold exceeding 3250 N.

Summarizing, the WIT braking pedal is a revolution that can withstand the same stresses that a traditional metal pedal must endure to meet automotive quality standards. Plastic throttle pedals and clutches can be found in today’s vehicles, but until now, there has never been an affordable plastic brake pedal. The world’s first plastic brake pedal debuted in 2013 on the Porsche 918 [66]. Regardless, the technology of the material for this pedal was vastly different, as it was made of endless fiber organosheets. This requires a much more complicated process and higher costs, which can be assumed for the Porsche hypercar but not for an average car. The WIT brake pedal proposed in this research is made with inexpensive materials and technologies that can compete in cost with the traditional metal alternative while dramatically reducing weight. It could be introduced into more affordable vehicles.

A retractable system was also designed during the NG-PED project. BATZ is aware of the increasing level of automation in the automotive sector. There are cars on the market with an increasing level of autonomy, and one of the future trends involves fully autonomous vehicles with retractable parts [67]. This transition will not happen overnight, there are situations and terrains where technology cannot currently drive a car autonomously, but there are other cases where the roads are well-defined and in optimal conditions, allowing the vehicle to clearly recognize vertical and horizontal traffic signs and operate itself. The pedals may be redundant in these cases, so retracting them for more space and comfort could be an innovative idea.

A retractable pedal concept has been devised for this purpose and is shown in Figure 10. The pedal can be retracted in 5 s for greater comfort.

#### 4.2.5. Functional Safety

As mentioned above, a system like the one presented in this article must be functionally safe [68]. The intention of this project is not to comply with ISO 26262 directly but to anticipate the requirements that will be needed for future projects. By acknowledging this, BATZ can position itself strategically. The following paragraphs will explain how the hardware was designed to be safe and redundant.

Firstly, two sensors of different natures are being used. The angle sensor is used to read the position of the pedal. Since the mechanical motion has a well-defined closed range, the inductive sensor enables the determination of the pedal’s position. Simultaneously, the force sensor detects the pressure exerted on the pedal. The first sensor requires movement, while the second does not.

This hardware architecture allows the ECU to detect the malfunction of the system, which could result in a reduction in the vehicle’s functionalities through a “safe state” or, in the worst case, an emergency stop. Both sensors must be linearized, and their measurement outputs should be correlated. Afterward, an analysis of hypothetical malfunctions is conducted.

The first scenario is shown in Figure 11, in a hypothetical case where the pedal gets stuck. The pedal system readings differ for the same scenario because the sensors read different physical magnitudes. The force sensor will provide a reading of the force the driver applies, even though the displacement will be shorter due to the stuck pedal. On the other hand, the inductive sensor will only read the area of the pedal system where the pedal is not stuck. In this example, there is a free movement between 0° and 21°, so the sensor will only measure in that range. Above that position, the pedal is stuck.

Since the reading from both sensors must be correlated, the vehicle’s ECU should recognize the different inputs and identify the problem. In this scenario, safety measures can be applied by third parties.

Another scenario could be that the main spring of the pedal has broken. Although this is extremely rare, it can be detected.

The example in Figure 12 shows the behavior of both sensors when the main spring is broken. In this scenario, the force sensor loses its preload, so the driver must exert an additional force at the beginning of the motion. On the other hand, at the end of the travel, there is no spring opposing resistance to the stroke, so the sensor curve for the same distance will increase earlier than under normal circumstances. Failure can be detected because the pedal is depressed at rest. No force is detected at rest, but the inductive angle sensor measures the pedal at an intermediate position. As mentioned above, if the ECU reads different values, it detects incoherence and will apply its own policy. More cases were identified in a broader functional safety analysis, and the conclusion is that they are detectable.

Continuing with functional safety and redundancy, using different communication protocols helps to increase fault tolerance. This prototype used SENT and analog output to transmit the measurement data. As mentioned earlier, the SENT protocol has a CRC to detect errors in the transmission of digital data, while it can also include additional information. On the other hand, non-digital signals provide continuous and smooth variations, allowing for precise representation of the measured parameters without the need for digital encoding or decoding. This enables direct and real-time interpretation of the sensor outputs, providing immediate feedback and response in the control system, although they are more noise-sensitive. In this case, redundancy allows the ECU to read the information from different paths. If some vehicle communication buses fail or stop working, another one will be available to obtain the information from.

The sensors must be powered from different sources to make the system fully redundant. This is a way to ensure the continuity of service because the probability of a failure affecting two power supplies simultaneously is very low. If the same power supply powers both sensors, both will be affected if it fails.

In a system such as the eBrake, functional safety is extremely important. Everything in a vehicle related to braking is very sensitive, and the analysis must be thoughtful to design a safe product. Adhering to standards such as ISO 26262 will help to mitigate any failures and the consequences of a potential failure.

### 4.3. Envisioning the Future

The NG-PED project has been a real asset to BATZ and has helped in exploring modern technologies and mechatronic designs. As mentioned above, the new BBW technology allows for completely different and innovative pedal designs, such as pedals with minimal or even zero displacement.

Although rare, such pedals on a vehicle are not new. In 1955, Citroën implemented a hydraulic brake with 10 mm of travel as a button on its DS model. In this car, the brakes reacted according to the force with which the driver pressed the pedal rather than the travel distance, which allowed for a more responsive and immediate braking experience. This brake was hydro-pneumatic; the system took advantage of the suspension’s high-pressure hydraulic system. The French manufacturer continued to use this mushroom-shaped pedal on the 1970 Citroën SM [69], which at the time was claimed by *Popular Science* magazine to have the shortest braking distance of any car ever benchmarked [70].

Except for the example above and other early automotive brakes that were hand-operated, most brake pedals feature a straightforward design, with a flat or slightly curved surface for the driver to step on. The pedal is usually mounted on the vehicle’s floor, attached to the dash panel, and coupled to the braking system by mechanical linkages. When the driver depresses the brake pedal, these linkages convert the pedal motion into hydraulic pressure used to operate the brakes. With the advent of electronics, the hydraulic braking system, or at least its mechanical connection to the pedal, can be eliminated. This also allows the pedal’s shape to be redefined, saving space and weight in the vehicle.

The project presented in this article has positioned BATZ at the forefront of design, resulting in innovative mechatronic prototypes. Figure 13 shows a demonstration of an electronic low-travel pedal. Next to the pedal, electronics have been implemented to read the force and displacement signals measured by the pedal’s sensors and display them as text on an LCD screen.

## 5. Conclusions

The significance of electronics in the automotive industry’s innovation is evident as most vehicle systems have undergone or are transforming. The article has reviewed the introduction of BBW technology in the vehicle, and with this trend, it is unlikely that the brake pedal will be left behind. Incorporating new functions and features is essential to prevent the commoditization of this key automotive component. The situation becomes much more complex by integrating sensors and electronic safety systems into the pedal box. It is no longer a rotating mechanical lever but a sophisticated mechanism where electronic development is the most important part. Consequently, research in this area is necessary to identify opportunities to add value and differentiate the electronic brake pedal from the traditional one. This will improve the emergency brake response, optimize regenerative braking in hybrid and electric vehicles, achieve better vehicle dynamics, and make the pedal modular; and with the new recycled plastic materials, it can be lightweight and ecological.

The NG-PED project is BATZ’s effort to create a brake pedal that is unique and innovative, moving away from being the brake pedal as a commodity. This research has resulted in the creation of different functional prototypes of an electronic and adjustable brake pedal, completely BBW in nature. Additionally, research was conducted in several areas, with the following outcomes:Sensors: Multiple prototypes of a new inductive angle sensor have been successfully produced, and the magnetic sensor has undergone improvements, including a low-power consumption sleep mode rated at 0.1 mA. Printed-ink strain gauge technology was investigated to optimize cost-effectiveness without sacrificing accuracy and dependability, and several prototypes were manufactured. The study focused on the resistivity dispersion of printed inks deposited on PCBs and their linearity. Despite a high resistivity dispersion among the samples, exceptional linearity was observed.Feel simulator: The OEMs have expressed their preferences for the force curve with respect to the user’s pedal stroke in the feel simulator. Two systems that satisfy the desired curve have been suggested and can be adjusted either passively or actively.Shape-memory alloy: The investigation indicates that utilizing SMA wire for the actively adjustable brake pedal produced favorable results. Through an electronic control system, it becomes a customizable feel simulator, which allows the driver to select from three distinct pedal stiffness levels when braking. While the functionality tests demonstrated satisfactory results, concerns have been raised regarding the suitability of these materials for the automotive industry. Despite the effectiveness of this solution, it must meet the strict requirements that the automotive sector demands.Water injection technique: The WIT technique has expanded its range of uses to brake pedals, producing eco-friendly pedals that support environmental sustainability. The technology maintains the pedal’s safety and structural integrity while reducing weight, aligning with the industry’s goals of enhancing fuel efficiency and reducing emissions. This novel brake pedal has undergone rigorous tests, achieving the same high standards as conventional metal pedals, including exceptional performance in the collapse test. This research marks a groundbreaking achievement by introducing the first affordable plastic brake pedal manufactured using WIT.Retractable pedal: This feature introduces a visionary concept to optimize space utilization within the driver compartment in future autonomous vehicles. This novel solution enhances the overall driving experience by offering the ability to retract the pedals when unused. It aligns with the evolving needs of autonomous driving scenarios and makes trips more comfortable.Low travel pedal: Currently, BBW technology is in its early stages of deployment. In response to this development, there are two options available. The first is to maintain the conventional design of the pedal, while the other is to explore new shapes and configurations. The decision will ultimately depend on the OEMs’ innovation level and drivers’ willingness to embrace changes. With BBW architecture, the pedal is no longer bound to a mechanical element, allowing for greater flexibility in its shape and placement. Furthermore, it can streamline the manufacturing process and costs for creating pedals suitable for left-handed driving (LHD) and right-handed driving (RHD). This prototype’s unique design deviates from the traditional pedal shape with low traveling.

Our research encompasses various areas and has resulted in significant scientific contributions. In the field of sensors, we have successfully developed multiple prototypes of a new inductive angle sensor, optimized magnetic sensors, and explored printed-ink strain gauge technology, achieving exceptional linearity despite resistivity dispersion. This technology is promising, and if it is suitable for the automotive industry, it will offer compact, cost-effective designs. The investigation into the feel simulator led to the development of two systems that meet the desired force curve requirements. Additionally, we explored using shape-memory alloy (SMA) wire for an actively adjustable brake pedal, offering customizable pedal stiffness levels. Further evaluation is needed for automotive adoption. We extended the water injection technique (WIT) to produce eco-friendly brake pedals that reduce weight by almost half while maintaining safety and structural integrity. Lastly, we introduced a visionary concept of a retractable pedal, optimizing space utilization in autonomous vehicles and enhancing driver comfort. This is an innovative solution and aligns with the increasing autonomy levels of vehicles.

This research explored the potential of brake-by-wire (BBW) technology and its implications for the future of automotive design. The developed prototypes demonstrate the feasibility and potential benefits of incorporating new functions into the brake pedal, highlighting the need for further research. Electronics will be the key to innovation in brake pedals, and BBW technology will play a significant role in this transformation. This article also focuses on the importance of functional safety and the upcoming automotive standards. Notably, when the NG-PED project started, there was no reference in the eBrakes market; as mentioned before, there is still none in production. In the meantime, however, various OEMs and competitors have started to show interest in these systems, which indicates that there has been hidden work while BATZ was developing its own solutions.

The conducted study has provided valuable knowledge, competitive advantages, and a comprehensive understanding of the current technologies and regulations that are essential for creating safe and dependable automotive components. The emergence of electronic braking pedal technology is a significant development in the automotive industry, aligned with the increasing trend of electronics in vehicles. Like electronic throttle control, the brake pedal also advances in the same direction, offering improved maneuverability, stability, and safety benefits. This project has positioned BATZ as an innovative leader in this field and instills confidence in OEMs regarding its proficiency in this technology.

## Figures and Tables

**Figure 1 sensors-23-06345-f001:**
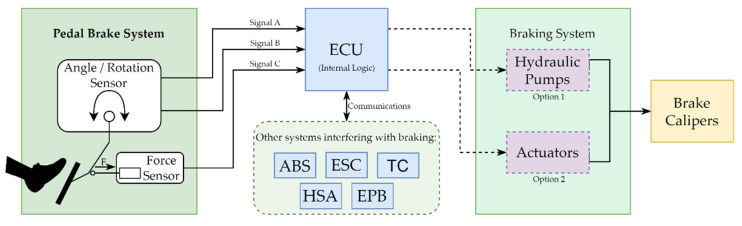
Brake-by-wire pedal hardware architecture. The received brake signals will control the brake system but may also interact with other safety systems.

**Figure 2 sensors-23-06345-f002:**
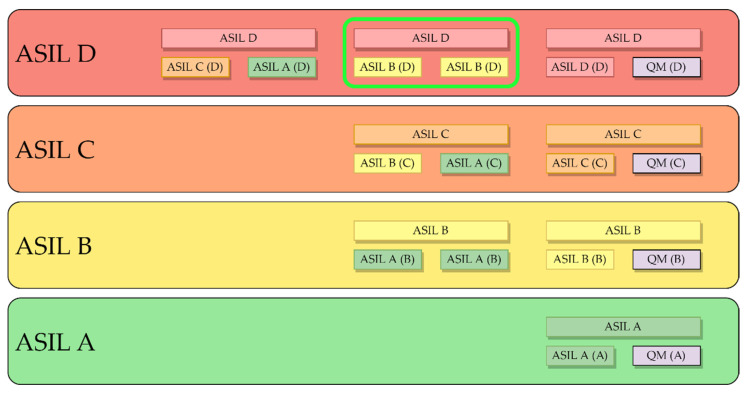
ASIL decomposition. Two ASIL B redundant systems can achieve ASIL D.

**Figure 3 sensors-23-06345-f003:**
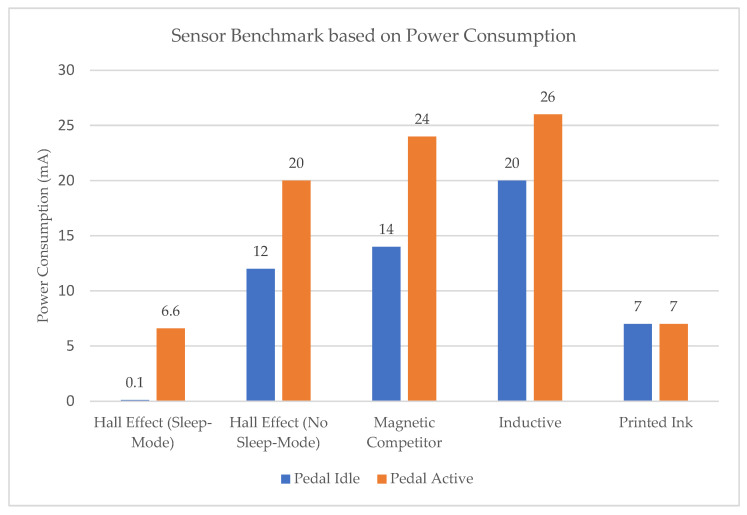
Sensor benchmark based on power consumption (the lower, the better).

**Figure 4 sensors-23-06345-f004:**
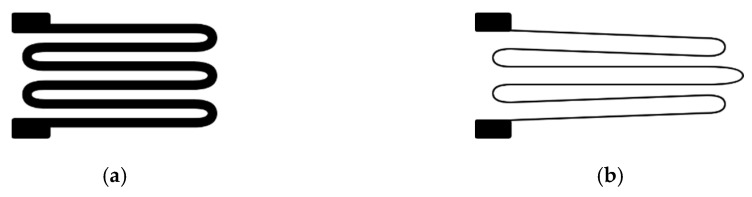
Diagram showing the physical behavior of a strain gauge. When a force is applied, it stretches, increasing its electrical resistance. (**a**) Strain gauge at rest position, broader tracks and better conductivity; (**b**) strain gauge in a stressed position, narrower tracks increase electrical resistance and decrease conductivity.

**Figure 5 sensors-23-06345-f005:**
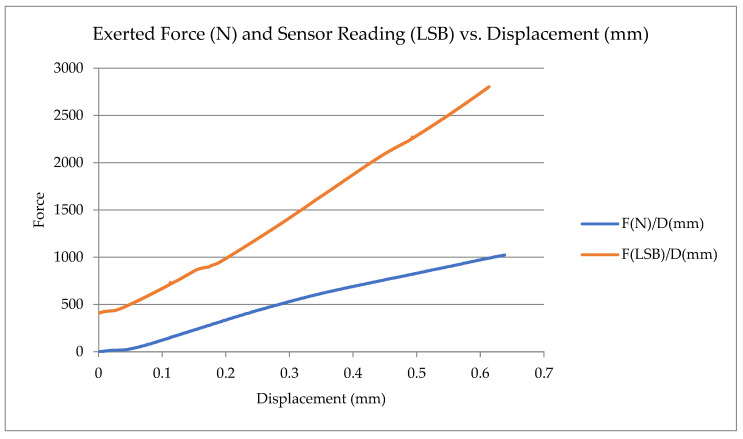
Exerted force and sensor reading versus displacement: in blue, the applied force in Newtons (N) and the physical displacement (mm); in orange, the digital signal value (LSB) transmitted by the sensor.

**Figure 6 sensors-23-06345-f006:**
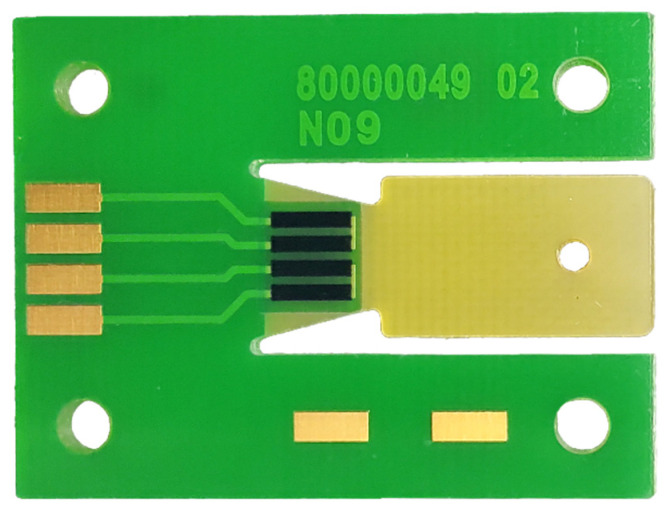
Prototype of a strain gauge based on printed-ink technology developed by BATZ. The conditioning stage does not appear on this PCB.

**Figure 7 sensors-23-06345-f007:**
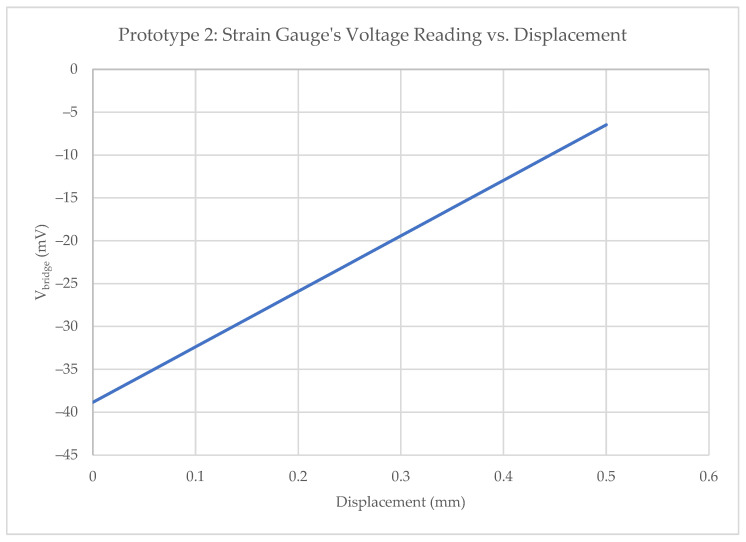
Prototype 2 results. A linear response is observed when measuring the voltage across the Wheatstone bridge as the gauge displacement increases.

**Figure 8 sensors-23-06345-f008:**
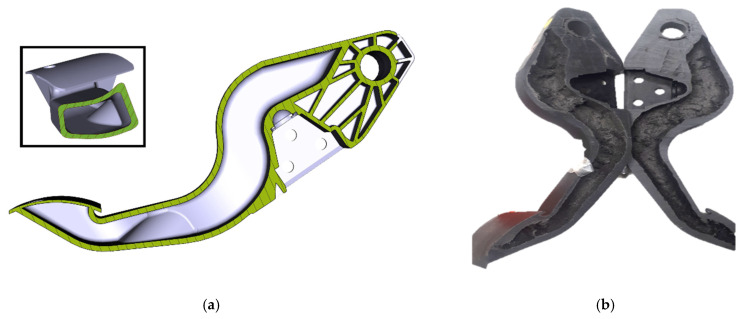
WIT prototype manufactured at BATZ. (**a**) View of the computerized 3D model. Note that the pedal’s interior is hollow; (**b**) the manufactured pedal was cut in half to analyze the internal structure.

**Figure 9 sensors-23-06345-f009:**
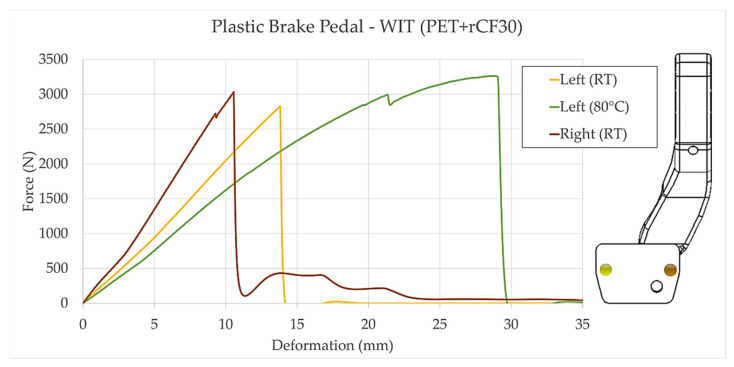
Collapse test results: room temperature (RT) left and right; 80 °C (Left).

**Figure 10 sensors-23-06345-f010:**
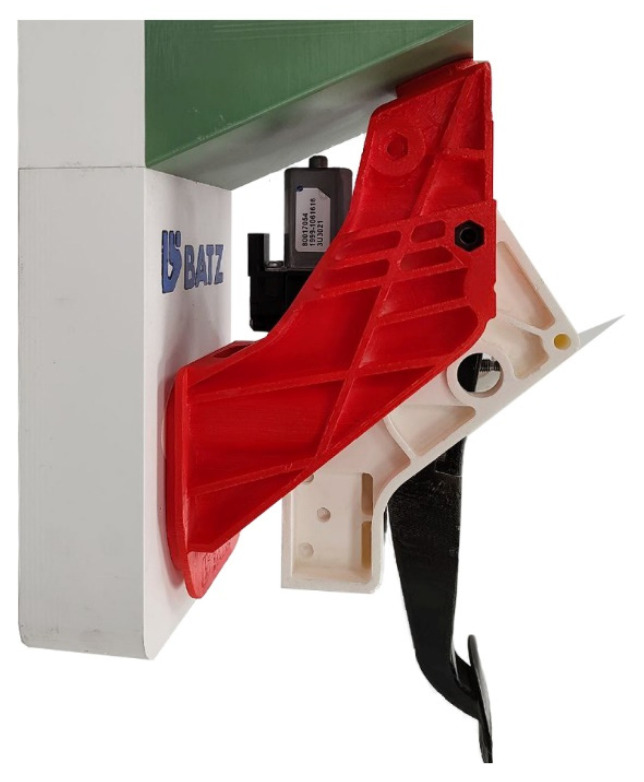
Retractable prototype. An actuator allows the pedal to retract to a flat position in 5 s.

**Figure 11 sensors-23-06345-f011:**
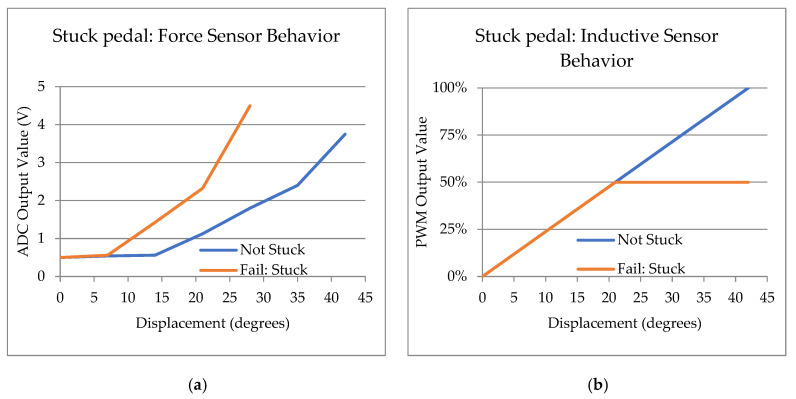
Behavior of the sensors in a hypothetical situation where the pedal is stuck halfway. (**a**) The curve of the force sensor will accentuate and increase before reaching the end of the stroke; (**b**) the travel sensor curve will be linear until it reaches the stuck position and will not be able to reach the end of travel.

**Figure 12 sensors-23-06345-f012:**
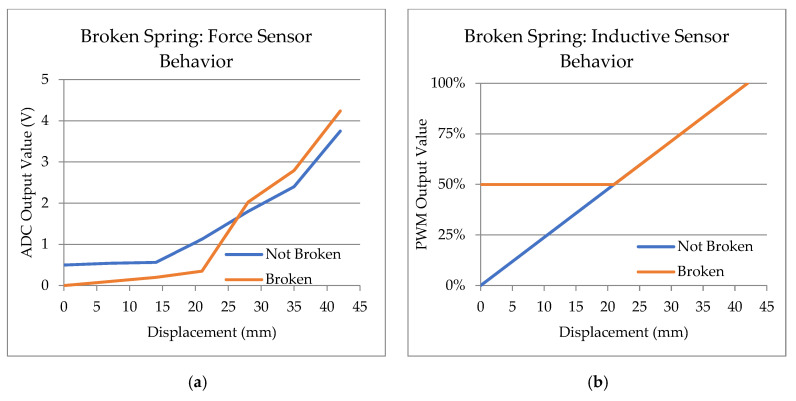
Behavior of the sensors in a hypothetical situation where the pedal’s main spring breaks. (**a**) The force sensor curve is slightly flat at the beginning of the stroke. In the middle, there is a noticeable increase in hardness, causing the force curve to rise rapidly and resemble the curve observed during normal operation. (**b**) In the absence of an operating spring, the pedal will naturally be positioned further forward. In this case, the travel sensor will detect and measure this specific condition, resulting in a reduced measurement range compared to its normal operating range.

**Figure 13 sensors-23-06345-f013:**
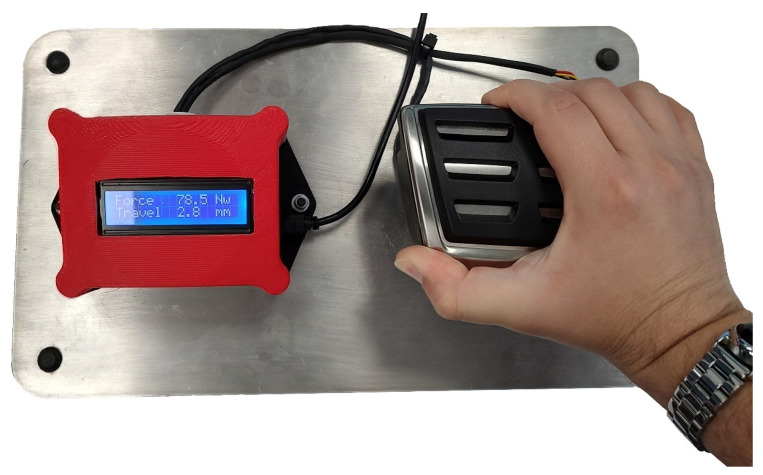
Electronic braking pad prototype. Force is applied to the pedal, and sensor readings are updated on the LCD.

**Table 1 sensors-23-06345-t001:** List of vehicle manufacturers and their models equipped with electronically controlled braking systems and the respective suppliers.

Supplier	System	OEM	Vehicles and Production Date
ADVICS	Electronic Controlled Brake (ECB)	Toyota	Previa/Estima Hybrid (2001), Alphard (2002), Prius (2004–2015)
		Lexus	RX 400h (2005–2009)
Hitachi	Electrically-Driven Intelligent Brake (EDIB)	Nissan	Fuga Hybrid (2010–2022), Leaf (2010–2017)
		Infiniti	M35h (2011–2019)
Continental	Regenerative Braking System (RBS)	Ford	Escape Hybrid (2005–2009), Fusion Hybrid (2009–2012)
		BMW	X6 Active Hybrid (2010–2012)
	MK C1	Alfa Romeo	Giulia (2015-Present), Stelvio (2016-Present)
		Audi	e-tron (2020–Present)
Bosch	Sensotronic	Mercedes-Benz	E-Class W211/S211 (2002–2010), CLS C219 (2004–2010), SL R230 (2001–2011), SLR R199 (2003–2009), Maybach W240 (2002–2013)
	iBooster	Volkswagen	e-Up! (2014–2016), e-Golf (2014–2021), Golf VII GTE (2017–2020)
		Opel	Ampera (2017)
		Porsche	918 Spyder (2013–2015)
	Hydraulic Apply System (HAS)	Renault	Zoe (2013–Present)
Mando	Active Hydraulic Boost (HAB)	Kia	Niro (2019–Present)

**Table 2 sensors-23-06345-t002:** Partial hazard analysis.

Hazard ID	Function	Hazard	Op. Mode	Hazardous Event	Consequence
H.1	Pedal system must be fail-safe.	Loss of communication.	Normal	Braking intention not processed.	Accident. Could imply fatalities.
H.2		Wrong braking order.	Normal	Unintended braking or missed braking intention.	
H.3		Loss of electric power.	Stop Normal	Failure in the power supply.	
H.4	Pedal must return to idle when not pressed.	Pedal got stuck.	Normal	Unintended braking.	Could provoke an accident.

**Table 3 sensors-23-06345-t003:** Partial ASIL determination and safety goals.

Hazard ID	Severity		Exposure		Controllability		ASIL	Safety Goal
H.1	Could provoke life-threatening injuries.	S3	Very low probability of connection loss in a point-to-point architecture.	E1	Difficult. The driver does not have control over data transmission.	C3	A	Pedal position must always be available.
H.2	Could provoke life-threatening injuries.	S3	The sensor’s fault rate is low.	E2	Difficult. The driver does not have control over the brakes in this situation.	C3	B	
H.3	Could provoke life-threatening injuries.	S3	Power supplies‘ fault rate is low.	E2	Uncontrollable if there is only one sensor and no backup power supply.	C3	B	Guarantee that the pedals are powered and available.
H.4	Could provoke an accident.	S3	Low probability of the pedal getting stuck.	E2	Controllable. Driver could actuate to avoid this hazard.	C2	A	Pedal must not get stuck. Stuck pedal should be detectable.

**Table 4 sensors-23-06345-t004:** Resistors values obtained from printed gauges.

	Prototype 1	Prototype 2	Prototype 3
Resistor A	120 k Ω	100 k Ω	121 k Ω
Resistor B	96 k Ω	108 k Ω	110 k Ω
Resistor C	98 k Ω	103 k Ω	105 k Ω
Resistor D	121 k Ω	108 k Ω	117 k Ω

## Data Availability

Not applicable.

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
