# Peer review of "Next-Generation Pedal: Integration of Sensors in a Braking Pedal for a Full Brake-by-Wire System"

_sensors, 2023, doi:10.3390/s23146345_

Round 1

Reviewer 1 Report (Previous Reviewer 1)

This release details the interesting studies and tests conducted in a better and clearer way, operating a rewiew similar to that of a patent report.

Author Response

Dear Reviewer,

Thank you for your feedback. I appreciate your positive comments and I'm pleased to hear that the revised version of the manuscript has met your expectations. Your support is greatly appreciated.

I have attached the letter I prepared for all the reviewers, which includes all the changes made based on the received suggestions.

Thank you once again for your time and effort in reviewing our manuscript.

Best regards,

Jose Ángel Gumiel

Reviewer 2 Report (Previous Reviewer 3)

Submit a clear version,two many typos or delete lines in the current version.

Moderate

Author Response

Dear Reviewer,

Thank you for your feedback on the manuscript. I appreciate your suggestions and have taken them into consideration during the revision process.

To address the issues raised, I have thoroughly reviewed the entire manuscript and made improvements. I have carefully corrected all the identified typos and inconsistencies, resulting in a clearer and more polished version. Additionally, I have focused on enhancing the overall readability of the text by making it more concise and coherent.

In addition, I have updated some references, ensuring their relevance and accuracy. I have included recent sources and removed an obsolete reference.

I would like to inform you that I have incorporated the performed modifications directly into the manuscript using the Microsoft Word comment feature. Additionally, I am attaching a separate response letter that includes a detailed summary of all the changes made, providing a comprehensive overview of the revisions.

I believe that these changes have significantly improved the quality and clarity of the manuscript. I hope that the revised version meets your expectations.

Thank you for your time and valuable feedback.

Best regards,

Jose Ángel Gumiel

Reviewer 3 Report (New Reviewer)

This article presents a fully electronic braking pedal which is an important actuator for the vehicle system. The work is interesting. Some minor revisions should be made to improve the quality of the paper.

1) Full brake-by-wire system is also used as the platform for autonomous driving. One of the reasons is that the response of the actuator is fast. As a result, a lot of work for the vehicle experiment test is based on the full brake-by-wire system: autonomous vehicle kinematics and dynamics synthesis for sideslip angle estimation based on consensus kalman filter, improved vehicle localization using on-board sensors and vehicle lateral velocity, automated vehicle sideslip angle estimation considering signal measurement characteristic, estimation on imu yaw misalignment by fusing information of automotive onboard sensors. It would be meaningful to include the above work. Based on the above applications, the application and prospect of wire-controlled actuators are very broad.

2) In the intervention logic design process of ABS and ESC, it is very important to obtain vehicle speed and acceleration with high precision. Currently, architectures based on multi-sensor fusion help to improve the robustness and accuracy of vehicle speed estimation. An automated driving systems data acquisition and analytics platform, yolov5-tassel: detecting tassels in rgb uav imagery with improved yolov5 based on transfer learning. By introducing and including the above work, it is helpful for readers to have a deeper understanding of the working principles of wire-controlled actuators such as ABS and ESC.

3) The title font in the figures is too large, please match the font size of the main text.

Author Response

Dear Reviewer,

Thank you for your valuable feedback on the manuscript. I appreciate your suggestions and have made significant efforts to address your concerns and improve the overall quality of the article.

Regarding the references, I have carefully reviewed the two references you shared with me and found them relevant to our study. I have incorporated them into the manuscript, ensuring their proper citation and integration within the context of our research.

In response to your comment about the font size in the figures, I have updated all the figures to match the font size of the main text, ensuring consistency throughout the document. This modification enhances the visual coherence and readability of the figures, aligning them with the overall manuscript style.

To enhance the introduction, I have provided additional background information and incorporated relevant references to establish a stronger foundation for our study. I have also revised the state of the art section, updating references and removing a obsolete source, thereby ensuring that our article reflects the most current research in the field.

Moreover, I have taken your suggestion seriously and worked on presenting the results more clearly. I have modified certain paragraphs and restructured the information to improve the overall readability and comprehension of our findings. These changes aim to provide a clearer understanding of our research outcomes and their significance.

I would like to inform you that I have included the performed modifications directly in the manuscript using the Microsoft Word comment feature. Additionally, I have attached a separate response letter that summarizes all the changes made, addressing your specific suggestions and incorporating the feedback from the other reviewers.

I believe that these revisions have significantly improved the manuscript.. I appreciate your feedback, as it has contributed to the refinement of our work.

Thank you once again for your valuable input.

Best regards,

Jose Ángel Gumiel

Round 2

Reviewer 2 Report (Previous Reviewer 3)

The scientific contribution should be highlighted, it is a technical report under the current version. Quality and figures should be improved seriously.

Quality of English have been improved.

Author Response

Dear Reviewer 2,

I am writing to provide an update on the revisions made to the manuscript based on your feedback and suggestions. Your comments have played a significant role in improving the paper's overall quality and scientific contributions.

Firstly, significant efforts have been made to enhance the quality of the figures. All images have been updated with higher resolutions, and Figures 6 and 13, which were previously too dark, have been retaken to ensure clarity and visual representation. Furthermore, Figure 8 (a) has been modified to include a different depiction, providing a more comprehensive illustration.

New relevant and up-to-date references have been included throughout the manuscript to strengthen the scientific content. The Abstract, Introduction, and Conclusions sections have undergone revisions to highlight the significant scientific contributions of the research. Particularly, the Conclusions section has been extensively modified to enhance clarity and emphasize the key findings.

I am grateful for bringing attention to the English grammar and sentence structure issues. Comprehensive revisions have been made to address these concerns, ensuring that the manuscript meets the high-quality standards expected by the journal.

The modifications mentioned above have successfully addressed the issues raised in your review. The manuscript now presents improved figures with enhanced resolution, updated and relevant references, and refined language and grammar. I am confident that these revisions have significantly strengthened the paper's scientific contributions and overall quality.

Once again, I would like to thank you for your review and feedback which has contributed to the enhancement of this manuscript. I trust the revisions will meet your expectations and align with the journal’s standards.

If you have any additional comments or suggestions, please don't hesitate to inform me. I appreciate your time and consideration.

Sincerely,

Jose Ángel Gumiel

This manuscript is a resubmission of an earlier submission. The following is a list of the peer review reports and author responses from that submission.

Round 1

Reviewer 1 Report

Interesting paper that deals with an original topic, we await the continuation whith the test on the veichles.

Reviewer 2 Report

The paper has briefly explained the history of BBW in the automotive industry, and how the Next Generation Pedal (NG-PED) developed by BATZ is gaining relevance in an industry context that advocates electronic innovation. On the other hand, it is also intended to study new materials and to apply new weight reduction techniques. All this has resulted in a fully functional and safe pedal prototype.

But this paper does not carry out theoretical research and detailed test verification on integration of sensors in a Braking Pedal. The author need to highlight this paper's innovative contributions. 

The quality of the English language is decent.

Reviewer 3 Report

See attached.
